# The Role of CXCL16 in the Pathogenesis of Cancer and Other Diseases

**DOI:** 10.3390/ijms22073490

**Published:** 2021-03-28

**Authors:** Jan Korbecki, Karolina Bajdak-Rusinek, Patrycja Kupnicka, Patrycja Kapczuk, Donata Simińska, Dariusz Chlubek, Irena Baranowska-Bosiacka

**Affiliations:** 1Department of Biochemistry and Medical Chemistry, Powstańców Wielkopolskich 72 Av., Pomeranian Medical University in Szczecin, 70-111 Szczecin, Poland; jan.korbecki@onet.eu (J.K.); patrycjakupnicka@o2.pl (P.K.); patrycja.kapczuk@pum.edu.pl (P.K.); d.siminska391@gmail.com (D.S.); dchlubek@pum.edu.pl (D.C.); 2Department of Medical Genetics, School of Medicine in Katowice, Medical University of Silesia, Medyków 18, 40-752 Katowice, Poland; kbajdak-rusinek@sum.edu.pl

**Keywords:** CXCL16, CXCR6, cancer, tumor, NKT cells, SR-PSOX, ADAM10, tumor microenvironment, inflammation

## Abstract

CXCL16 is a chemotactic cytokine belonging to the α-chemokine subfamily. It plays a significant role in the progression of cancer, as well as the course of atherosclerosis, renal fibrosis, and non-alcoholic fatty liver disease (NAFLD). Since there has been no review paper discussing the importance of this chemokine in various diseases, we have collected all available knowledge about CXCL16 in this review. In the first part of the paper, we discuss background information about CXCL16 and its receptor, CXCR6. Next, we focus on the importance of CXCL16 in a variety of diseases, with an emphasis on cancer. We discuss the role of CXCL16 in tumor cell proliferation, migration, invasion, and metastasis. Next, we describe the role of CXCL16 in the tumor microenvironment, including involvement in angiogenesis, and its significance in tumor-associated cells (cancer associated fibroblasts (CAF), microglia, tumor-associated macrophages (TAM), tumor-associated neutrophils (TAN), mesenchymal stem cells (MSC), myeloid suppressor cells (MDSC), and regulatory T cells (T_reg_)). Finally, we focus on the antitumor properties of CXCL16, which are mainly caused by natural killer T (NKT) cells. At the end of the article, we summarize the importance of CXCL16 in cancer therapy.

## 1. Introduction

A tumor mass comprises not just cancer cells but also tumor growth supporting cells and tumor-suppressing immune cells [1]. All these cells forming the tumor microenvironment secrete various factors into the intercellular space. To date, many of these factors have been characterized. The most significant of these are chemokines, a group of approximately 50 chemotactic cytokines [2]. Earlier studies have characterized chemokines as a significant factor in immune system cell function [3,4]. However, over time, more studies have emerged showing the significance of a variety of chemokines in cancer development and growth [5]. One such chemokine is CXC motif chemokine ligand 16 (CXCL16), belonging to the sub-family of α-chemokines. To date, there has not been a review summarizing current knowledge on CXCL16 and its receptor CXC motif chemokine receptor 6 (CXCR6) in a tumor. Therefore, the aim of this review is to collect and summarize all available knowledge about CXCL16 to make it more accessible to scientists studying the role of chemokines in cancer.

## 2. CXC Motif Chemokine Ligand 16 (CXCL16): Background Information

CXCL16 is a chemokine distinct from other CXC chemokines. The *CXCL16* gene is located on chromosome 17p13 [6], separate from other chemokine genes, and it has poor homology with the other chemokines [7]. There are two CXCL16 transcripts, 1.8 kb and 2.5 kb in length, formed by alternative splicing [8]. Both transcripts differ in the 3′-noncoding regions and the location of their expression. The 1.8 kb transcript is found mostly in the spleen, thymus, and testis. In contrast, the 2.5 kb transcript is found in the heart, kidney, liver, lung, peripheral blood leukocytes, pancreas, and prostate [8]. Alternative RNA splicing produces two other transcripts having 70 (CXCL16v1) and 121 (CXCL16v2) extra nucleotides, as demonstrated in dendritic cells [9]. In these additional sequences, a STOP codon is present. This results in a shorter protein having only the chemokine domain. The other domains typically found in CXCL16 are not present.

After translation, a hydrophobic signal peptide is cleaved at the N-terminus of the CXCL16 polypeptide [7]. The human CXCL16 protein is 254 aa long [6,7], while the murine CXCL16 is 246 aa long and is 44% similar to the human CXCL16 [8]. The newly synthesized CXCL16 for both species is the so-called membrane-bound form of CXCL16 (mCXCL16) [6]. The structure of this protein is very similar to transmembrane CX3CL1 [10] because it consists of a small (24–27 aa long) cytoplasmic tail with a YXPV motif [6]. This motif can be phosphorylated on tyrosine to provide an SH2-binding site. mCXCL16 also consists of a CXC chemokine domain and a transmembrane domain (Figure 1) [6]. Both of these domains are separated from each other by a spacer region, approximately 100 aa long, rich in serine, threonine, and proline. It is a site of heavy O-glycosylation which results in the formation of a mucin-like stalk [6,7], essential for chemokine domain presentation and, therefore, significant for mCXCL16 properties [11,12]. mCXCL16 has a molecular weight of 60 kDa.

After CXCL16 cleavage, the 35 kDa chemokine domain is released and becomes a soluble form of CXCL16 (sCXCL16). This process is regulated by disintegrin and metalloproteinase 10 (ADAM10) [13,14,15]. However, disintegrin and metalloproteinase 17 (ADAM17) may also be responsible for sCXCL16 shedding in the absence of ADAM10 [16] or by using phorbol 12-myristate 13-acetate (PMA) as an inflammatory inducer [13]. Studies on mesangial cells have shown that pro-inflammatory cytokines increase sCXCL16 shedding via ADAM10 and ADAM17 [17]. Although sCXCL16 is released from mCXCL16, the remainder of the mCXCL16 (mucin-like stalk, transmembrane domain, and cytoplasmic tail) is still present in the cell membrane. These domains are firstly proteolytically cleaved by γ-secretases and then proteolytically degraded [15].

Importantly, many papers do not distinguish between two forms of CXCL16. Researchers just alter the expression of the CXCL16 gene or change the expression of two CXCL16 forms at the same time; then they test selected parameters of the experiment. In those cases, we report the findings as referring to the unspecified “CXCL16” which may denote both forms at the same time. When writing about mCXCL16 and sCXCL16, we mean the specific form of CXCL16.

CXCL16 is classified as an α sub-family chemokine because it has a CXC motif at the N-terminus [7]. Nevertheless, this chemokine lacks the ELR motif, which is common in other pro-angiogenic CXC chemokines [6,7]. CXCL16 is expressed in lymphoid organs such as the thymus, spleen, lymph nodes, and Peyer’s patches, but not in bone marrow. It is also expressed in the liver, lung, small intestine, and kidney [6,7]. Moreover, CXCL16 is highly expressed in the epidermis, where it is produced by keratinocytes [18]. mCXCL16 is expressed in macrophages [8,19] and is also present in splenic and lymph node dendritic cells (DC), blood myeloid DC, and monocyte-derived DC. Interestingly, its expression increases after DC maturation and after exposure to pro-inflammatory factors [6,19]. It has been also shown that CXCL16 is expressed on CD19^+^ B cells [7]. 

Both forms of CXCL16 have different functions. sCXCL16 is a chemokine that is responsible for the chemotaxis of cells bearing the CXCR6 receptor [6,7] while mCXCL16 is a transmembrane protein. mCXCL16 may have adhesion protein properties that bind to CXCR6 (Table 1) [12,20]. However, signal transduction from CXCR6 via mCXCL16→CXCR6 is not necessary for cell adhesion. This property of mCXCL16 is important in immune cell accumulation at an inflammation site due to increased mCXCL16 expression on vascular walls by pro-inflammatory cytokines [20]. mCXCL16 also mediates the adhesion of Gram-negative and Gram-positive bacteria [11]. This leads to bacterial phagocytosis by cells expressing mCXCL16, such as macrophages and DC [8,11]. mCXCL16 may also act as a receptor that causes signal transduction into the cell. This signal can be induced by sCXCL16 [21] as well as by CXCR6 [22]. In the first case, this effect is called inverse signaling; in the second case–reverse signaling. Additionally, signal transduction from mCXCL16 is insensitive to pertussis toxin [23], and so it has a different mechanism than signaling mediated by the G protein-coupled receptor (CXCR6). The sCXCL16-induced activation of mCXCL16 causes the activation of extracellular signal-regulated kinase (ERK) mitogen-activated protein kinase (MAPK) [21,23] and Akt/protein kinase B (PKB) pathways [21]. This leads to increased proliferation and formation of apoptosis-resistant tumor cells [21,23]. In glioblastoma or melanoma cells, CXCR6 activates the mCXCL16→ERK MAPK pathway, which causes migration but not proliferation of tumor cells [22].

CXCL16 expression is increased by pro-inflammatory cytokines. This is important for the accumulation of immune cells at the inflammatory reaction sites, but the influence of the various cytokines is cell-specific. In ectopic endometrial stromal cells, tumor necrosis factor *α* (TNF-α) increases CXCL16 expression [24]. In vascular smooth muscle cells, interferon-γ (IFN-γ) increases the expression of this chemokine, but TNF-α does not have such an effect [34]. In human umbilical vein endothelial cells (HUVEC), TNF-α, IFN-γ, interleukin (IL)-1β, and IL-6 increase the expression of CXCL16 [20]. Research on keratinocytes has shown that TNF-α, IFN-γ and peptidoglycan increase the expression of CXCL16 in these cells [18]. Also, the expression of CXCL16 is increased by ionizing radiation, which is important during radiotherapy [35,36]. Another factor that increases the expression of CXCL16 is hypoxia and hypoxia-inducible factor-1 (HIF-1) [37,38].

## 3. CXCR6: Background Information

Apart from mCXCL16, we know only one another receptor for sCXCL16, namely CXCR6, also known as STRL33 and TYMSTR [6,7]. The highest expression of CXCR6 is observed in the appendix, lymph node, placenta, spleen, and thymus [39]. CXCR6 is also expressed in immune system cells, in particular, naïve CD8^+^ cells [6], CD3^−^CD16^−/low^CD56^+^ and CD3^−^CD16^low^CD56^−^ natural killer (NK) cells [40], natural killer T (NKT) cells [6], activated CD4^+^ and CD8^+^ T cells [6,41], and in 30–40% of γδ T cells [41]. 

The expression of CXCR6 increases in T cells during cell differentiation by DC [42]. CXCR6 is mainly expressed on T helper 1 (Th1) or T cytotoxic 1 (Tc1) cells [42]. Some types of T cell also express CXCR6 [42]. It has been shown that most of the cells expressing CXCR6 lack at least one homing lymphoid organ receptor [42], indicating that CXCR6 expression is a marker of immune cell differentiation. Due to CXCL16 expression on dendric cells, the CXCL16→CXCR6 axis has an important role in the interaction of immune cells with DC [6,11]. CXCR6 can be also found on CD19^+^ B cells [40] while other studies have shown that B cells do not express this receptor [6,41]. CXCR6 is not expressed in vivo on DC, macrophages, monocytes or neutrophils [6,41]. However, in vitro research on macrophages differentiated by granulocyte-macrophage colony-stimulating factor (GM-CSF) and macrophage colony-stimulating factor (M-CSF) has shown that CXCR6 expression is higher in the M1 than in the M2 macrophage subset [43]. This shows that macrophages can have CXCR6 expression if they are polarized by appropriate factors. However, more research is required in this area because although many factors can differentiate and influence macrophage differentiation, their influence on CXCR6 expression has not been studied.

The CXCL16→CXCR6 axis is not only important for the physiology of immune system cells but also in brain development, by participating in the migration of glial precursor cells [44]. 

CXCR6 requires Gα_i_ for signal transduction and, therefore, is sensitive to pertussis toxin [12,40]. Activated CXCR6 leads to the activation of several signaling pathways, one causing calcium mobilization (Figure 2) [7,12,25]. This, in response to sCXCL16, mediates the chemotaxis of the cells with activated CXCR6. Importantly, in all other 7-transmembrane chemokine receptors, there is a DRY motif that is essential for receptor activation and signaling transduction to G proteins. In CXCR6, instead of a DRY motif, a DRF motif is present [25]. Nevertheless, it does not cause changes in the activation of the phosphatidylinositol-4,5-bisphosphate 3-kinase (PI3K)→Akt/PKB axis and ligand binding, receptor internalization, or receptor recycling, but only increases cell adhesion to mCXCL16. Also, this motif change reduces calcium signaling activated by CXCR6, which is associated with a reduction in chemotactic response to sCXCL16 [25]. This effect seems to be cell-specific [25,45].

The second pathway activated by CXCR6 is the PI3K→Akt/PKB axis [46]. This pathway is also sensitive to pertussis toxin [26]. The PI3K→Akt/PKB pathway activates other proteins and pathways, for example, nuclear factor κB (NF-κB) which leads to TNF-α expression [26,27]. As a consequence of NF-κB activation, proliferation and cell migration increase [26,28]. Likewise, Akt/PKB activates the mammalian target of rapamycin (mTOR) which is associated with increased proliferation and proangiogenic factor expression [30]. Akt/PKB may also activate forkhead box O3a (FOXO3a) [30]. CXCR6 activation via Akt/PKB increases the action of adenosine diphosphate (ADP) on platelets, which causes platelet adhesion and platelet activation [46]. 

A third pathway activated by CXCR6 is ERK MAPK [24,31]. This leads to the activation of RhoA and F-actin formation and consequently to tumor cell migration [31]. 

CXCL16 can also activate p38 MAPK in dermal microvascular endothelial cells and endothelial progenitor cells, although it has not been confirmed whether this effect depends on CXCR6 expression [47]. Experiments on hepatocellular carcinoma SK-HEP-1 and HCCLM3 cells did show that the activation of CXCR6 causes a reduction in p38 MAPK activation and thus the activation of glycogen synthase kinase 3β (GSK3β), which leads to a reduction in the amount of β-catenin in the plasma membrane [32].

## 4. The Importance of CXCL16 in Non-Neoplastic Diseases

CXCL16 is involved in the pathogenesis of atherosclerosis. Inflammatory cytokines such as TNF-α, IFN-γ, IL-1β and IL-6 increase the expression of CXCL16 on endothelial cells, as shown in a study on HUVEC [20]. This leads to the adhesion of THP-1 monocytes, which are human monocytic leukaemia cells. A study on peripheral blood mononuclear cells (PBMC) shows that monocytes do not express CXCR6 [41], which means that in the pathogenesis of atherosclerosis, monocytes may not be directly recruited through the CXCL16→CXCR6 axis. In principle, this effect may be indirect by recruiting CXCR6^+^ T cells [48]. CXCL16 plays an important role in the later stages of atherosclerosis. A high expression of CXCL16 is present in macrophages in the intima of atherosclerotic lesions but not in normal aortas [49]. mCXCL16 has scavenger receptor activity and can be a receptor for oxidized lipoproteins. For this reason, it was first named the scavenger receptor for phosphatidylserine and oxidized lipoprotein (SR-PSOX) [49]. As such, it can participate in the uptake of oxidized low-density lipoprotein (oxLDL) by macrophages [49] and vascular smooth muscle cells [34]. CXCL16 can also cause proliferation of aortic smooth muscle cells and increase the expression of TNF-α in these cells [26], which is associated with atherosclerotic vascular disease. A study in low-density lipoprotein (LDL) receptor-deficient mice showed that CXCL16 activity is athero-protective [50]. For this reason, more research is needed on the role of CXCL16 in the pathogenesis of atherosclerosis.

Another situation where CXCL16 plays a significant role is in liver diseases, in particular, nonalcoholic fatty liver disease (NAFLD). During inflammatory reactions, the expressions of CXCL16, CXCR6, and ADAM10 increase in the liver [51]. In NAFLD, hepatocytes produce CXCL16 [52] which activates hepatic stellate cells that produce collagen and transform into myofibroblasts [52]. Interestingly, activation of CXCR6 on hepatocytes reduces inflammation, fibrosis, and the death of these cells, as demonstrated in mice with an activated NF-κB pathway [53]. Despite this, CXCL16 increases lipid accumulation, extracellular matrix (ECM) excretion and reactive oxygen species (ROS) production in hepatocytes [51]. CXCL16 is also responsible for the accumulation of NKT cells in the liver [53,54,55,56]. These cells participate in inflammatory reactions, leading to liver fibrosis progression. Also, during septic shock, the expression of CXCL16 is increased in the liver [57]. This leads to the recruitment of activated T cells and consequently to endotoxin-induced lethal liver injury. Importantly, the NK and NKT cells are not responsible for septic shock symptoms in the liver.

Transplantology is another issue where CXCL16 plays an important role. During allogeneic heart transplantation in mice, the CXCL16→CXCR6 axis is involved in the recruitment of NKT cells into the transplanted organ [58], which is related to allograft tolerance. Nevertheless, CXCL16 can also cause graft-vs-host disease (GvHD) due to its participation in the recruitment of activated CD8^+^ T cells into the liver, leading to GvHD-induced hepatitis [59]. 

The CXCL16→CXCR6 axis is also involved in fibrosis in various organs. For example, during inflammatory reactions, the expression of CXCL16 increases on tubular epithelial cells [60]. This leads to the recruitment of bone marrow-derived fibroblast precursors, cells expressing CXCR6 [60,61], which participate in the pathogenesis of renal fibrosis. CXCL16 is also engaged in pulmonary fibrosis [30], which was demonstrated in human pulmonary fibroblasts (MRC-5 cell line), where this chemokine caused an increase in proliferation and collagen production. This was associated with the activation of the forkhead box O3a (FOXO3a) via the CXCR6→Akt/PKB pathway [30].

CXCL16 is also essential in the homeostasis of intestinal defense. It is responsible for the distribution of intestinal group 3 innate lymphoid cell (ILC3) subsets and IL-22 production [62]. This leads to an increase in antimicrobial peptide expression, which is involved in the defense against bacteria. CXCL16 can also be taken as a marker of inflammatory bowel disease (IBD) [63]. It seems that CXCR6^+^CD4^+^ T cells do not show colitogenic properties, in contrast to CXCR6^−^CD4^+^ T cells [64]. However, in inflamed colonic tissues, CXCL16 expression occurs on macrophages where it participates in the Th1 immune response [63]. 

The CXCL16→CXCR6 axis also participates in the development of endometriosis [24]. In ectopic endometrial stromal cells, TNF-α increases the expression of CXCL16. This causes migration and invasion of these cells, which is associated with the development of endometriosis. 

CXCR6 is also an entry cofactor for human immunodeficiency virus (HIV)-1 [6,39,65,66] and HIV-2 [67] and so may be significant in the course of HIV infection. This is also supported by the association of polymorphisms in the CXCR6 gene with the control of viraemic disease [68,69]—its expression is downregulated in this disease [70]. In particular, the presence of specific polymorphisms in CXCR6 leads to rapid progression of acquired immunodeficiency syndrome (AIDS) [71] and influences the effectiveness of highly active antiretroviral therapy (HAART) [72]. Also, CXCL16 levels increase in HIV infection as the virus increases the release of this chemokine by macrophages [73]. This is associated with disease progression as CXCL16 increases HIV replication. 

## 5. Effect of the CXCL16→CXCR6 Axis on Tumor Cells

### 5.1. Regulation of CXCL16 Expression in Tumors

CXCL16 is produced by the cancer cells of tumors such as glioblastoma multiforme [74], lung cancer [75,76,77], lymphoma [78], and nasopharyngeal carcinoma [79,80]. According to data from the TCGA PanCancer Atlas Studies (https://www.cancer.gov/tcga, accessed on 15 January 2021) developed by “The cBioPortal for Cancer Genomics” (https://www.cbioportal.org, accessed on 15 January 2021), cancers are characterized by low levels of mutation in the *CXCL16* and *CXCR6* genes [81,82]. It is estimated that nearly 8% of stomach adenocarcinomas have a deletion of the *CXCL16* gene. In contrast, the level of *CXCL16* gene mutations in other tumors is lower, at about 3% of tumors, such as for deletion, amplification, or mutation. The CXCR6 gene also does not undergo frequent mutations in tumors. In Diffuse Large B-Cell Lymphoma, more than 4% of these tumors have a deletion in the CXCR6 gene [81,82]. The expression of this chemokine is increased by the activation of Notch1 [80] and ERK MAPK [79] in nasopharyngeal carcinoma cells. The expression of CXCL16 and CXCR6 is also increased by the inflammatory response, in particular, by pro-inflammatory cytokines (TNF-α and IFN-γ) in prostate cancer cells [83]. For this reason, CXCL16 is considered an inflammation marker associated with cancer [83]. Also, CXCL16 expression in HUVEC is increased by lipopolysaccharide (LPS) and NF-κB activation [84]. 

Another factor increasing the expression of CXCL16 is hepatitis C virus (HCV) infection in hepatocytes, which plays a significant role in liver cancer development [85]. Osteopontin (OPN) increases the expression of CXCR6 in hepatocellular carcinoma cells [86]. In pancreatic ductal adenocarcinoma there is a loss of somatostatin receptor subtype 2 (SSTR2) expression [87]. This leads to the activation of PI3K and a consequent increase in CXCL16 expression. A loop is formed in which CXCL16 activates PI3K and thus its own synthesis. This effect is significant in promoting the initiation and progression of this cancer. CXCL16 expression also depends on receptors, e.g., complement C5a receptor 1 (C5aR1) in lung cancer cells [88]. 

The CXCL16→CXCR6 axis is also influenced by microRNAs. In particular, the expression of CXCL16 is downregulated by miR-451 [89] and miR-873-5p [90]. miR-361-5p reduces the expression of CXCR6 [91]. Due to the pro-tumorigenic properties of CXCL16, these microRNAs are downregulated in tumors. This was confirmed by research on osteosarcoma cancer and miR-451 [89], papillary thyroid cancer and miR-873-5p [90], and hepatocellular carcinoma and miR-361-5p [91].

However, in some types of cancer cells, expression of CXCL16 is often downregulated, e.g., by promoter methylation in renal cell carcinoma cells [92]. Also in breast cancer cells, high CXCL16 expression is found in less aggressive cell lines [93]. Expression of CXCL16 can be also found in tumor-associated cells such as microglial and endothelial cells in glioblastoma multiforme [74], fibroblasts, endothelial cells, and macrophages in non-small cell lung cancer [76], mesenchymal stem cells (MSC) in gastric cancer [94,95], myeloid-derived suppressor cells (MDSC) in breast cancer [96], and cancer-associated fibroblasts (CAF) in breast cancer brain metastasis [97].

### 5.2. Effect of CXCL16 on Cancer Cell Proliferation

CXCL16 influences the intensity of tumor cell proliferation, depending on the type of tumor as well as the form of CXCL16. sCXCL16 increases proliferation of colorectal cancer HT-29 cells [98], gastric cancer MKN45 cells [94,95], glioblastoma U343 cells [21], glioblastoma GL261 cells [99], hepatocellular carcinoma HepG2 and Hep3B cells [91], melanoma IGR37 and IGR39 cells [100], meningioma cells [23], non-small cell lung cancer NCI-H2126 and NCI-H520 cells [77], prostate cancer PC3 cells [83], and schwannoma cells [101]. Also, increased CXCL16 expression enhances the proliferation of papillary thyroid cancer TPC-1 and K-1 cells [90]. Likewise, CXCR6 expression in gastric cancer HGC-27 cells [102] is positively associated with the proliferation of cancer cells, as well as CXCL16 expression in osteosarcoma U2OS and SaOS2 cells [89], and CXCR6 in osteosarcoma MG-63 cells [103]. 

The mechanism of sCXCL16 influence on proliferation depends on the type of tumor cell. In schwannoma cells, sCXCL16 increases proliferation via the ERK MAPK signaling pathway [101]. In gastric cancer, this effect is related to STAT3 activation, which leads to an increase in Ror1 receptor tyrosine kinase expression [95]. In contrast, in osteosarcoma MG-63 cells, the influence of the CXCL16→CXCR6 axis on proliferation depends on Akt/PKB activation [103]. Liang et al. showed that silencing of CXCL16 expression in A549 and PC-9 cells leads to a decrease in cell proliferation via the reduction of NF-κB activation [28]. In meningioma cells [23] and glioblastoma U343 cell line [21], the increase in proliferation depends on inverse signaling by activating mCXCL16.

The role of the CXCL16→CXCR6 axis may be closely related to cancer stem cells (CSC). CXCR6 is expressed in more aggressive CSC and is a marker of CSC asymmetric self-renewal division, as shown in melanoma cells [100,104]. Also in glioblastoma multiforme, it has been shown that CXCR6 is expressed predominantly on the CSC [74,99]. However, the exact relevance of CXCL16 on CSC requires further research. 

sCXCL16 does not affect the proliferation of tumor cells such as the Hodgkin lymphoma cell line L428 [78], diffuse large B-cell lymphoma OCI-LY8, and OCI-LY10 cells [105], pancreatic ductal adenocarcinoma BxPC3, CAPAN-1, COLO-357, and T3M4 cells [106]. Also, in hepatocellular carcinoma SK-HEP-1 and the HCCLM3 cells [32], changes in CXCR6 expression do not affect tumor cell proliferation. A study on the hepatocellular carcinoma SMMC-7721 cell line showed that the downregulation of CXCR6 expression reduces cell proliferation [33]. However, in that study sCXCL16 was not tested–only the expression of CXCR6 was determined. In contrast, in MDA-MB-231 breast cancer cells, overexpression of CXCL16 did not affect proliferation [93]. 

Interestingly, CXCL16 may also reduce the proliferation of cancer cells, which was shown in renal cell carcinoma cells [92]. In gastrointestinal stromal tumor GIST-T1 and GIST882 cells, sCXCL16 reduces proliferation by activating CXCR6 and reducing ERK MAPK activity [107]. Also, reduced CXCL16 expression increases the proliferation of non-small cell lung cancer A549 and NCI-H460 cells [76], as well as prostate cancer DU145 and PC3 cells [108]. In breast cancer MDA-MB-231 cells, overexpression of CXCL16 causes apoptosis [93], while in diffuse large B-cell lymphoma, sCXCL16 promotes sensitivity to TNF-α-induced apoptosis [105]. This effect depends on NF-κB activation and increased TNF-α expression, as well as increased TNF-α secretion of sCXCL16 due to increased ADAM10 expression. sCXCL16 may also cause the accumulation of macrophages via NKT cells [109]. Macrophages release TNF-α which leads to the apoptosis of cancer cells as shown in colorectal cancer metastasis in the liver [109]. 

In some types of cancer cells, increased expression of CXCL16 may reduce proliferation [76,108]. It has been postulated that this effect is related to the increased expression of mCXCL16 in the plasma membrane and co-expression of CXCR6, which results in contact inhibition of tumor cell growth [76]. This mechanism may also explain the inhibition of tumor cell migration with the overexpression of CXCL16 [14,93].

### 5.3. Effect of CXCL16 on Cancer Cell Migration

sCXCL16 causes the migration and invasion of cancer cells, which has been shown on numerous cell models, such as breast cancer MDA-MB-231 cells [110], colorectal cancer HT-29 cells [98], gastric cancer AGS cells [111], gastric cancer MKN45 cells [95], glioblastoma multiforme GL261 cells [99], hepatocellular carcinoma HepG2 and Hep3B cells [91], lung cancer H292 cells [75], nasopharyngeal carcinoma cells [112], non-small cell lung cancer NCI-A549, 95D, H2126 and NCI-H520 cells [75,77], ovarian cancer SKOV-3 and OVCAR-3 cells [27,113], pancreatic ductal adenocarcinoma T3M4 and BxPC3 cells [106], papillary thyroid cancer BHP10-3 cells [114], prostate cancer LNCaP, PC3, and DU145 cells [115,116,117], and schwannoma cells [101]. CXCR6 expression is also positively associated with the migration of breast cancer MCF-7 and MDA-231 cells [31], gastric cancer HGC-27 cells [102], gastric cancer SGC-7901 cells [118], hepatocellular carcinoma SMMC-7721 cells [33], hepatocellular carcinoma SK-HEP-1 and HCCLM3 cells [32], osteosarcoma MG-63 cells [92], and prostate cancer PC3 and C4-2B cells [29]. 

The effect of CXCL16 on cancer cell migration may be increased by chronic hypoxia, which increases HIF-1-dependent CXCR6 expression and thus the sensitivity of tumor cells to sCXCL16, as shown in breast cancer MDA-MB-231 cells [110]. Nevertheless, not all cancer cells respond to sCXCL16. Studies on neuroblastoma HTLA-230 and GI-LI-N cells have shown that although they express CXCR6, this receptor is nonfunctional [119] and sCXCL16 does not cause their migration. 

Although sCXCL16 causes the migration of cancer cells, the mechanism differs depending on the selected research model. In ovarian cancer SKOV-3 cells [27] and papillary thyroid cancer BHP10-3 cells [114], migration depends on the PI3K→Akt/PKB pathway. In prostate cancer, PC3 and C4-2B cell migration depends on the CXCR6→PI3K→Akt/PKB→mTOR pathway activation [29]. A study on prostate cancer LNCaP and PC3 cells showed that due to the PI3K→Akt/PKB pathway and protein kinase C (PKC), phosphorylation of Ezrin takes place along with an increase in F-Actin formation, which leads to the migration of cancer cells [117]. Also in these cells, activation of PI3K→Akt/PKB and FAK pathways causes α_v_β_3_ integrin clustering which leads to increased cell adhesion to bone marrow endothelial cells and consequently to bone metastasis [117]. A reduction in CXCL16 expression in A549 and PC-9 cells decreases tumor cell migration by reducing NF-κB activity [28]. This proves the importance of this transcription factor in the migration of tumor cells.

The CXCL16→CXCR6 axis also causes migration via other pathways. Breast cancer MCF-7 and MDA-MB-231 cell migration is caused by ERK MAPK activation [31]. Activation of this cascade causes signaling via RhoA kinase, which inhibits the effect of cofilin, which, in turn, inhibits F-actin formation. Taken together, this means that ERK MAPK activation leads to F-actin formation and tumor cell migration [31]. On the other hand, in gastric cancer, migration is associated with CXCL16 activation of STAT3, which leads to an increase in Ror1 receptor tyrosine kinase expression [95]. Furthermore, the CXCL16→CXCR6 axis reduces p38 MAPK activity in hepatocellular carcinoma SK-HEP-1 and HCCLM3 cells, which leads to the activation of GSK3β and consequently, to a decreased amount of β-catenin in the plasma membrane [48]. This increases the migration of cancer cells. Also, the CXCL16→CXCR6 axis causes epithelial–mesenchymal transition (EMT) as confirmed in colorectal HT-29 cancer cells [98] and osteosarcoma MG-63 cells [103]. In the latter, this effect depends on CXCR6 activation of the Akt/PKB pathway and the increased effect of TGF-β on the EMT process. 

Activation of CXCR6 by CXCL16 leads to increased expression and secretion of matrix metalloproteinases (MMP) in many types of cancer cells, such as breast cancer cells [110], gastric cancer cells [118], hepatocellular carcinoma cells [120], non-small cell lung cancer cells [77], in highly invasive ovarian cancer cells (not in less invasive cells) [113], papillary thyroid cancer cells [90], and prostate cancer cells [29,115,117]. In particular, there is an increase in the expression, secretion, and activity of MMP-1, MMP-2, MMP-3, MMP-8, MMP-9, MMP-11, MMP-13, and MMP-14. MMP-1, MMP-8, and MMP-13 belong to the category of collagenases. MMP-2 and MMP-9 belong to gelatinases, MMP-3 and MMP-11 to stromelysins, and MMP-14 to membrane-type MMP [121]. All these enzymes, except for MMP-11, degrade collagens, gelatin [122] and other ECM proteins, as well as participate in the activation or degradation of various factors important in cancer progression: chemokines, inflammatory cytokines, anti-inflammatory cytokines, adhesive proteins, growth factors, and other MMP [122]. By degrading the ECM and releasing various factors from the degraded ECM, MMP cause the migration and invasion of cancer cells [121]. MMP can also cause angiogenesis by releasing pro-angiogenic factors [123,124]. CXCL16 may also inhibit the migration of tumor cells. Upregulation of CXCL16 in renal cell carcinoma ACHN3 cells [14], breast cancer MDA-MB-231 cells [93], prostate cancer DU145 and PC3 cells [108] reduces the migration of these cells. This is due to the co-expression of mCXCL16 with CXCR6. mCXCL16 is an adhesion protein to its CXCR6 receptor [12,20]. Co-expression of these two proteins on the same cell causes the adhesion of cells with each other and leads to difficulties in the migration of these cells. Activation of CXCR6 can also inhibit cell migration, in particular, in regulatory T cells (T_reg_), activation of CXCR6 by sCXCL16 at a concentration below 0.3 ng/mL causes migration, but above 0.3 ng/mL inhibits the migration of these cells [107]. Also in gastrointestinal stromal tumor GIST-T1 and GIST882 cells, CXCR6-dependent sCXCL16 expression inhibits migration and EMT in cancer cells [107].

### 5.4. Effect of CXCL16 on Metastasis

CXCL16 plays a significant role in tumor metastasis. Studies on patients show that CXCR6 expression is greater in metastasis than in primary tumors of such cancers as cervical cancer [125], Ewing sarcoma family tumor [126], gastric cancer [102,118], melanomas [127], nasopharyngeal carcinoma [112], ovarian carcinoma [128], papillary thyroid cancer [114], and prostate cancer [116]. This shows that CXCR6 is significant in the formation of metastases. 

Among others, the axis CXCL16→CXCR6 is significant in bone metastasis, since CXCL16 is highly expressed in bone [126] by bone marrow stromal cells [129] and osteocytes [115]. For this reason, circulating cancer cells that express CXCR6 may be retained in the bones. Activation of CXCR6 causes α_v_β_3_ integrin clustering and hence an increase in tumor cell adhesion to bone marrow endothelial cells, as shown in prostate cancer cells [117]. CXCL16 also has an osteoclastogenic activity which is necessary for the formation of bone metastasis by lung cancer A549M1 cells [88]. 

It is also postulated that the CXCL16→CXCR6 axis is involved in breast cancer brain metastasis due to high CXCL16 production by brain metastatic CAF [97]. Also due to the high expression of CXCL16 in the liver and lung, it is postulated that the CXCL16→CXCR6 axis participates in metastasis in these organs [8,112,116,126]. CXCR6 expression has been associated with the lymph node metastasis of many tumors [102,112,114,118,125,128], with some exceptions. Prostate cancer studies have shown lower CXCR6 expression in lymph node metastases than in the primary tumor [116]. Also, the CXCL16→CXCR6 axis is required in the peritoneal metastasis of ovarian cancer [130]. TGF-β1, secreted by a tumor, causes fibrosis of the peritoneal mesothelial cells. These cells increase the expression of CXCL16, which then aids in the implantation of ovarian cancer cells and the formation of peritoneal metastasis [130].

## 6. Role of CXCL16→CXCR6 Crosstalk on the Tumor Microenvironment

### 6.1. Effect of CXCL16 on Angiogenesis and the Role of Hypoxia on CXCL16 Function

The CXCL16→CXCR6 axis also plays an important role in tumor angiogenesis (Figure 3). Its effect may be direct or indirect. sCXCL16 causes vascular endothelial growth factor (VEGF)-independent capillary tube formation, and proliferation of endothelial cells, as shown in dermal microvascular endothelial cells, human endothelial progenitor cells [47], and HUVEC [38,96,131]. In HUVEC, this process depends on the activation of PI3K→Akt/PKB, p38 MAPK and ERK MAPK [38,96,131]. Also, sCXCL16 causes chemotaxis of HUVEC, but this effect is not dependent on ERK MAPK [38,131]. 

The CXCL16→CXCR6 axis may also indirectly induce angiogenesis. This axis increases the expression of VEGF in human hepatocellular carcinoma SMMC-7721 cells [33], HUVEC [38], and prostate cancer PC3 and C4-2B cells [29]. Also, CXCL16 increases CXC motif chemokine ligand 8 (CXCL8)/IL-8 expression in prostate PC3 and C4-2B cancer cells [29] and in hepatocellular carcinoma SK-HEP-1 and HCCLM3 cells [32]. It is known that in prostate cancer cells this effect depends on activation of the CXCR6→Akt/PKB→mTOR pathway [29]. CXCL8/IL-8 is a proangiogenic factor [132] and CXCL16 may indirectly induce angiogenesis by increasing the expression of this chemokine.

The role of CXCL16 in HUVEC may be regulated by HIF-1. This chemokine, via ERK MAPK, p38 MAPK, and Akt/PKB, increases the level of HIF-1α under normoxia [38]. This also leads to an increase in CXCL16 expression in these cells. CXCL16 activity may also occur in hypoxic regions, where the expression of this chemokine is upregulated by chronic hypoxia, as shown in hepatocellular carcinoma Huh-7 and HepG2 cells [37]. In HUVEC, CXCL16 expression is increased by HIF-1 [38]. For this reason, chronic hypoxia may increase the expression of CXCL16 in these cancer cells, although this still needs to be confirmed through further studies. Also, CXCR6 expression is increased by chronic hypoxia in breast cancer MDA-MB-231 cells and HUVEC [110], with this effect dependent on HIF-1 regulation [110]. In A549 and SPC-A1 pulmonary adenocarcinoma cells, chronic hypoxia does not alter the expression of CXCL16 [133].

### 6.2. CXCL16→CXCR6 and Tumor-Associated Cell Crosstalk

The involvement of CXCL16 in tumorigenesis is closely related to the tumor-associated cells on which it acts and from which this chemokine is secreted into the tumor microenvironment (Table 2, Figure 4).

#### 6.2.1. Cancer-Associated Fibroblasts

One source of CXCL16 in tumors are CAFs, as shown in triple-negative breast cancers [135] and non-small cell lung cancer [76]. CAFs are fibroblasts involved in carcinogenesis. They mainly participate in the pro-tumorigenic production of the ECM and secrete a variety of factors involved in cancer progression [150]. These cells, under the influence of factors secreted from the cancer cells, begin to secrete sCXCL16 in large amounts [135]. At the same time, factors secreted from the monocytes are also important in the induction of CXCL16 expression and, therefore, this process requires the participation of three types of cell: cancer cells, monocytes, and fibroblasts. This is important in triple-negative breast cancers, although not all cell lines of this type of cancer increase the expression of CXCL16 in fibroblasts [135].

#### 6.2.2. Endothelial Cells

The CXCL16 expression present in a tumor is also present in endothelial cells, as shown in glioblastoma multiforme [74,134], meningioma [136], and non-small cell lung cancer [76]. Inflammatory reactions, such as from pro-inflammatory cytokines, increase the expression of CXCL16 in these cells [20]. Endothelial cells secrete many factors, such as sCXCL16, that support carcinogenesis [120].

#### 6.2.3. Tumor-Associated Macrophages

CXCL16 is important for the functioning of tumor-associated macrophages (TAM)–monocytes polarized into macrophages by factors found in the tumor microenvironment. TAM participate in many tumor processes, in particular, by increasing the proliferation and migration of cancer cells, causing cancer immune-evasion, and participating in angiogenesis [151]. As CXCL16 attracts monocytes [135,143], it participates in the recruitment of monocytes into the tumor niche, which are then differentiated into TAM. Studies on the expression of CXCR6 have shown that blood monocytes do not express this receptor [6,42], and so further studies are needed on the importance of sCXCL16 as a monocyte chemoattractant. 

The differentiation and polarization of TAM involve many factors. It has been shown that sCXCL16 may also be a macrophage polarizing factor [144]. Such polarized macrophages show features of the M2 macrophage subset: increased expression of CD163 and decreased expression of CD80, CD86, and HLA-DR. Also, such macrophages secrete large amounts of IL-10 and IL-15 which inhibit normal NK cell functioning [144]. Importantly, CXCL16 also causes microglia polarization into an anti-inflammatory phenotype [99]. Microglia are brain-resident macrophages that are important in glioma (brain tumors) maintenance and progression [152]. The effect of CXCL16 on microglia requires further research because in glioblastoma multiforme these cells have a very low expression of CXCR6 receptor [74,140]. Also, in the non-small cell lung cancer model, CXCR6 was not expressed in stromal cells such as fibroblasts, endothelial cells, and macrophages [76]. In Vitro polarized macrophages with M-CSF or GM-CSF have much higher CXCR6 expression than TAM in glioblastoma multiforme [43,140]. For this reason, more detailed studies imaging the expression of CXCR6 on TAM in other tumors are required. A significant source of sCXCL16 in the tumor is TAM, as shown in colorectal cancer [153], glioblastoma multiforme [74], meningioma [136], non-small cell lung cancer [76], papillary thyroid cancer [114,145], and hepatocellular carcinoma [146]. In glioblastoma multiforme [74] and meningioma [136], the microglia might also be a source of CXCL16. Research on rectal cancer has shown a decreased expression of CXCL16 in macrophages [154]. Expression of this chemokine in TAM is increased by factors secreted by cancer cells into the tumor microenvironment [114]. In hepatocellular carcinoma, this factor has been named colony-stimulating factor 1 (CSF1) [146]. Then, sCXCL16 is also secreted by TAM, which increases migration and proliferation of cancer cells. However, sCXCL16 secreted by TAM causes tumor infiltration by anti-cancer tumor-infiltrating lymphocytes (TIL) [153].

#### 6.2.4. Myeloid-Derived Suppressor Cells

The CXCL16→CXCR6 axis is also important in the functioning of MDSC in the tumor. As MDSC reduce the overly intense response from the immune system, they are responsible for cancer immune-evasion in cancer [155]. CXCL16 expression is increased in MDSC by factors secreted by cancer cells, as shown in mammary carcinoma 4T1 cells [96]. This process is important for the induction of angiogenesis in the tumor. CXCL16 also causes the survival of MDSC, which is important in the accumulation of these cells in the tumor niche [141]. However, the importance of CXCL16 may differ between different MDSC groups. CXCR6 expression occurs at a similar level between polymorphonuclear myeloid-derived suppressor cells (PMN-MDSC) and monocytic myeloid-derived suppressor cells (Mo-MDSC), although the latter show a higher expression of CXCL16 [142].

#### 6.2.5. Tumor-Associated Neutrophils

There are indications that CXCL16 is also involved in the functioning of tumor-associated neutrophils (TAN). These are cells derived from neutrophils, which, under the influence of the tumor microenvironment, transform into cells supporting tumorigenic processes [156]. Nevertheless, neutrophils in the blood do not express CXCR6 [6,41,147]. For this reason, they should not be recruited by sCXCL16 into the tumor niche. However, studies on hepatocellular carcinoma have shown that CXCR6 is important in recruiting TAN to the tumor niche [32]. Also, studies on pancreatic carcinoma show that TAN express CXCR6 [147]. Expression of this receptor may be dependent on low-grade chronic inflammation within the tumor. The inflammatory response, including TNF-α, increases the expression of CXCR6 in neutrophils [147]. Nevertheless, the exact role of the CXCL16→CXCR6 axis on TAN function still needs to be thoroughly investigated.

#### 6.2.6. Mesenchymal Stem Cells

CXCR6 is expressed on MSC [137]. For this reason, these cells are recruited by sCXCL16 into the tumor niche [137,138,139]. This chemokine then participates in the conversion of these cells into cancer-associated fibroblasts, as demonstrated in prostate cancer [137]. Also in gastric cancer, sCXCL16 is secreted by MSC [94,95], and this process begins after Wnt5a→Ror2 activation. Then, the expression of CXCL16 is increased into the tumor microenvironment. This leads to increased migration of gastric cancer cells. In contrast, in breast cancer there is a CXCL16-CXCL10 loop between breast cancer cells and MSC [139]. In this model, breast cancer cells secrete CXCL16 and MSC secrete CXCL10. This loop is important in the migration of cancer cells.

#### 6.2.7. Astrocytes

Interestingly, in brain tumors such as glioblastoma multiforme, CXCL16 is expressed in astrocytes [134]. The role of these cells in carcinogenesis processes orchestrated by CXCL16 requires further research.

#### 6.2.8. Regulatory T Cells

CXCL16 also participates in the recruitment and pro-tumorigenic functions of T_reg_. These are lymphocytes that mainly participate in cancer immune-evasion [157]. sCXCL16 is a chemotactic agent for T_reg_ [85]. In tumors such as nasopharyngeal carcinoma [148], renal cell carcinoma [149] T_reg_ show higher expression of CXCR6 compared to blood T_reg_. This shows that these cells can be recruited into the tumor niche by sCXCL16. sCXCL16 may also increase the growth of T_reg_ at a concentration below 0.3 ng/mL [107]. The effect of sCXCL16 depends on this chemokine concentration. Above this concentration, sCXCL16 reduces the growth of T_reg_ [107]. The half-maximal inhibitory concentration (IC) of CXCL16 for inducing the inhibition of an increase in T_reg_ is 6.57 ± 0.81 ng/mL [107]. Also at this concentration it reduces T_reg_ migration; this effect depends on CXCR6 activation.

#### 6.2.9. Anti-Cancer Tumor-Infiltrating Lymphocytes

sCXCL16 is also known as a chemotactic agent for anti-cancer TIL [40,153]. This is why an increased expression of this chemokine causes infiltration of tumor cells, which consequently has an anti-cancer effect [158]. In particular, sCXCL16 causes infiltration of the tumor by activated NK cells [159,160,161] and CD8^+^CXCR6^+^ T cells [35]. This process is especially important after radiotherapy followed by increased CXCL16 expression in tumor cells [35,160]. Also, CXCL16 increases proliferation of CD4^+^ T cells [83]. 

Nevertheless, the most important cells in the anti-cancer activity of CXCL16 are NKT cells. These are cells with NK and T cell characteristics, which might have anti-cancer or pro-cancer properties depending on the NKT subset [162]. For example, the CXCL16→CXCR6 axis is important in recruiting NKT cells into the lung [163]. In part, the process may be mast cell-dependent. Importantly, in the lung, the NKT cells act against cancer. 

sCXCL16 is essential in the migration of NKT cells [164,165] and CD4^+^ T cells [165] to the liver. However, Cullen et al. showed that CXCL16 may not play a key role in the migration of NKT cells to the liver, but just in activating these cells [166]. The migration of NK cell and T cells to the liver is sCXCL16-independent [164]. 

One of the factors controlling the liver expression of CXCL16 is the intestinal flora. At least in mice, the production of secondary bile acids in the gut depends on the presence of Gram-positive bacteria, in particular *Clostridium* spp. [167]. Primary bile acids increase the expression of CXCL16, while secondary bile acids reduce the expression of CXCL16 in liver sinusoidal endothelial cells [167]. 

If CXCL16 is highly expressed in a cancer cell, it increases the migration of anti-tumor lymphocytes to such cells [168]. In the liver, NKT cells and CD4^+^ T cells are important in the Th1 response. They cause the removal of senescent hepatocytes and act against cancer [167,168]. More precisely, they inhibit the formation of liver cancer and the formation of liver metastasis. In particular, these anti-cancer properties are associated with invariant natural killer T (iNKT) cells [169], which directly exhibit cytotoxic properties and also secrete IFN-γ, a cytokine that enhances the anti-cancer immune response [166,170]. NKT cells also cause liver infiltration by M1 macrophages [109] and act against cancer by producing and secreting TNF-α.

## 7. The CXCL16→CXCR6 Axis in Tumors

### 7.1. CXCL16

Compared to adjacent non-cancerous tissues or normal tissue, the expression of CXCL16 within tumors is elevated, for example in colon cancer [171], colorectal cancer [153,172], gastrointestinal stromal tumors [107], gastric carcinomas [173], glioblastoma multiforme [99,134,174], lung cancer [28], nasopharyngeal carcinoma [148], osteosarcoma [89], ovarian cancer [27,113,128], pancreatic ductal adenocarcinoma [87,106], papillary thyroid cancer [90,145], prostate cancer [116,175], and schwannomas [101]. However, increased expression of CXCL16 does not occur in all types of cancer. For example, in rectal cancer [154] there is a lower expression of CXCL16 than in normal tissue. In gastric cancer [102] CXCL16 expression does not differ from adjacent non-cancerous tissues. Upregulation of CXCL16 in many cancers is related to its role as an inflammation marker [83,134], and inflammation in tumors creates a microenvironment enhancing further growth of a tumor [176]. 

CXCL16 expression in tumor and lymph nodes correlates with disease development, in particular tumor size, stage, grade and metastasis [28,116,125,126,145,171,172,177,178,179]. However, this is not the case in all cancers. For example, in ovarian carcinoma [128] CXCL16 tumor expression is not associated with the clinical stage or lymph node metastasis. Whereas in gastric carcinomas [173], lower expression of CXCL16 in the tumor is associated with lymphatic invasion, and in renal cell carcinoma with the tumor stage [14]. 

Due to the increased expression of CXCL16 within the tumor development, a higher expression of this protein is associated with worse overall survival for a patient with cervical cancer (*p* = 0.089) [125], stage III/IV colorectal cancer [172], Ewing sarcoma family tumor [126], gastrointestinal stromal tumors [107], lung cancer [28], ovarian carcinoma [128,180] and prostate cancer (Table 3) [108]. Also, a higher expression of CXCL16 in regional lymph nodes is associated with poorer overall survival for colon cancer patients [171]. CXCL16 may also be a serum marker of worse overall survival for cancer patients with colorectal cancer [98], gastrointestinal stromal tumors [107], or ovarian cancer [180]. CXCL16 may be also a urine biomarker of urothelial carcinomas [178]. 

On the other hand, a higher expression of CXCL16 is associated with better overall survival in patients with colorectal cancer [153], gastric carcinoma [173], non-small cell lung cancer [76], and renal cell carcinoma [14]. In bladder cancer [177], ovarian cancer [180], and papillary thyroid cancer [145], the level of CXCL16 expression is not related to the patient’s overall survival. The tumor-type linkage of CXCL16 expression with the overall survival is also shown in data from “The Human Protein Atlas” (https://www.proteinatlas.org, accessed on 15 January 2021) [182,183]. There, a higher CXCL16 expression is associated with better overall survival for only 4 out of 17 types of cancer, and with worse overall survival for 4 out of the 17 (Table 4).

### 7.2. CXCR6

Analyses of CXCR6 expression show that the tumor has a higher expression compared to adjacent non-cancerous tissues, especially in tumors such as bladder cancer [177], breast cancer [31], gastric cancer [102,118], gastrointestinal stromal tumors [107], glioblastoma multiforme [134], hepatocellular carcinoma [32,33], non-small cell lung cancer [77], ovarian cancer [27,113,128], osteosarcoma [103], pancreatic ductal adenocarcinoma [87,106,184], prostate cancer [116,175], and schwannomas [101]. 

In colorectal adenocarcinomas [185] and gastric carcinomas [173], the expression of CXCR6 is lower than in normal tissue. A higher expression of CXCR6 is associated with the development of the tumor disease. In particular, with lymph metastasis in advanced clinical stage and grade [33,102,116,118,125,126,128,177]. Also, CXCR6 expression is greater in metastasis than in the primary tumor, such as in cervical cancer [125], Ewing sarcoma family tumor [126], gastric cancer [102,118], melanomas [127], nasopharyngeal carcinoma [112], ovarian carcinoma [128], papillary thyroid cancer [114], and prostate cancer [116]. This shows the possible mechanism of metastasis.

Similar to CXCL16, an elevated CXCR6 expression is also associated with poorer overall survival for patients with cervical cancer [125], clear cell renal cell carcinoma [186], Ewing sarcoma family tumor [126], gastric cancer [102], gastrointestinal stromal tumors [107], hepatocellular carcinoma (*p* = 0.064) [48], and prostate cancer [108] (Table 5). In the early stage of pancreatic ductal adenocarcinoma, a higher expression of CXCR6 is associated with better overall survival [184]. In bladder cancer [177], non-small cell lung cancer [179], ovarian cancer [128,180], and papillary thyroid cancer [14], the expression of CXCR6 is not associated with patient overall survival. The data published in “The Human Protein Atlas” (https://www.proteinatlas.org, accessed on 12 January 2021) do not confirm a clear negative impact of increased CXCR6 expression on tumor overall survival [182,183]. Here, in most cancer types, higher CXCR6 expression is associated with better overall survival (12 out of 17 types of cancer), while in only 2 out of 17 with worse overall survival.

## 8. CXCL16→CXCR6 Axis and Anti-Cancer Therapy

Studies on colorectal cancer [153] and nasopharyngeal carcinoma [148] have shown that the level of CXCL16 expression is associated with tumor infiltration by anti-cancer TIL, and so CXCL16 can be used in cancer therapy. An example of this is with ionizing radiation, as used in radiotherapy, which increases the expression of CXCL16 in the tumor cells, such as in colon carcinoma, breast carcinoma, fibrosarcoma, prostate carcinoma [35,160,187]. Also, instead of radiotherapy, an sCXCL16-conjugated antibody can be used [158]. It releases sCXCL16 via the cleavage of furin, an enzyme found on pancreatic cancer cells. This triggers the migration of anti-tumor TILs into the tumor, in particular, activated CD8^+^CXCR6^+^ T cells [35] and activated NK cells [158,160]. 

The aforementioned therapeutic methods can be combined with immunotherapy [187]. In the first step of therapy, there is an increase in the expression of CXCL16, a chemokine involved in tumor infiltration by anti-tumor TIL. Subsequently, immunotherapy is applied, and TIL specifically directed against cancer cells are introduced into the patient’s organism. These cells accumulate in a tumor where CXCL16 is up-regulated. 

On the other hand, CXCL16 also plays an important role in tumorigenic processes. For this reason, the use of CXCL16-neutralizing antibody [87] or an antagonist of the CXCR6 receptor [188] inhibits tumor progression in in vitro experiments, in particular in pancreatic ductal adenocarcinoma [87] and hepatocellular carcinoma [188]. 

CXCL16 plays an important role in side effects or resistance to anti-cancer drugs. The CXCL16→CXCR6 axis has been shown to be important in docetaxel resistance of prostate cancer cells [189]. It is related to CXCL16 activation of pro-survival pathways such as ERK MAPK and NF-κB. CXCL16 also participates in cisplatin treatment of acute kidney injury [190]. An increased expression of CXCL16 in renal tubular epithelial cells during this therapy leads to the infiltration into the kidneys by immune system cells, which causes inflammatory responses and apoptosis of the tubular epithelial cell. For this reason, it is postulated to use CXCL16-neutralizing antibody or CXCR6 inhibitors during cancer treatment with docetaxel or cisplatin in order to improve the applied therapies [189,190]. 

Along with other chemokines, CXCL16 is involved in the regulation of the cellular dormancy of glioblastoma multiforme cells after exposure to temozolomide (TMZ) [191], a standard drug used in the treatment of this cancer [192]. Exposure of glioblastoma multiforme cells to TMZ stops cell division. After discontinuation of the therapy, tumor cell proliferation resumes [191]. This process is regulated by the chemokine system, in particular CXCL16, and also by CX3C motif chemokine ligand 1 (CX3CL1) and CXC motif chemokine ligand 12 (CXCL12) [191]. More detailed studies on the effects of chemokines in cellular dormancy are required to develop an effective therapeutic approach.

CXCL16 may also be a marker of a patient’s response to cancer therapy, for example in the administration of bevacizumab to non-squamous non-small cell lung cancer patients. This monoclonal antibody against VEGF is used in solid tumor therapy [193]. Patients who had lower plasma levels of CXCL16 than before the treatment showed a longer overall survival [181]. This area of research requires more studies on patients with other cancers.

## 9. Conclusions: Perspectives for Future Research

The importance of the CXCL16→CXCR6 axis in tumorigenesis has been well established. CXCL16 is a marker of inflammatory reactions characteristic for cancer and is significant in the induction of proliferation and migration of neoplastic cells, intercellular communication in the tumor niche, angiogenesis, as well as in the recruitment and differentiation of various cells in the tumor niche. Nevertheless, little is known about the role of mCXCL16 in the tumor as all studies have focused on sCXCL16 or changes in *CXCL16* gene expression. These studies do not detail whether the effect observed is due to mCXCL16 or sCXCL16. Also, a poorly studied area of knowledge is the importance of this sCXCL16 in cell responses to chemotherapy. To date, only a few articles have addressed the importance of this chemokine for developing side effects or tumor resistance to anticancer drugs.

## Figures and Tables

**Figure 1 ijms-22-03490-f001:**
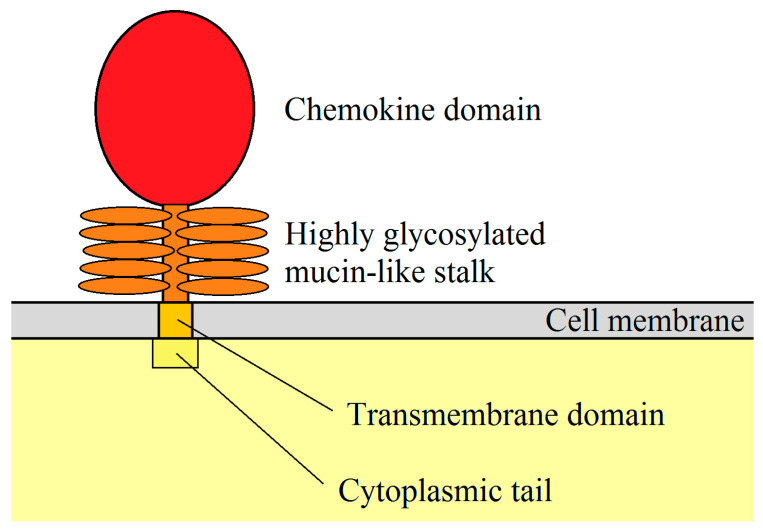
mCXCL16 structure. CXC motif chemokine ligand 16 (CXCL16) is a membrane protein consisting of a chemokine domain, highly glycosylated mucin-like stalk, transmembrane domain and cytoplasmic tail.

**Figure 2 ijms-22-03490-f002:**
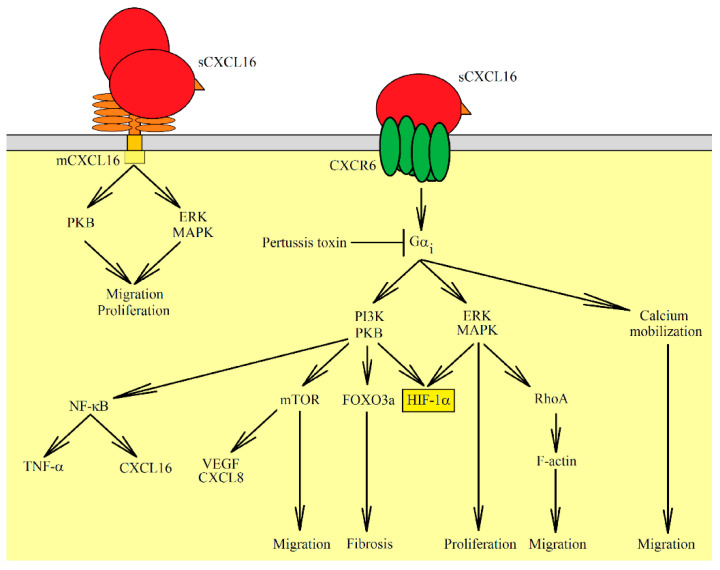
Signaling pathways activated by sCXCL16, a ligand for CXCR6. This chemokine can also interact with mCXCL16 leading to signal transduction. Through mCXCL16, the Akt/PKB and ERK MAPK pathways are activated which leads to cell migration and proliferation. Activation of CXCR6 causes signal transduction through three pathways: PI3K→Akt/PKB, ERK MAPK, and calcium mobilization. Activation of these pathways is sensitive to pertussis toxin, in contrast to signal transduction via mCXCL16. Activation of ERK MAPK causes cell proliferation and migration. Also, ERK MAPK and Akt/PKB cause HIF-1α phosphorylation, which increases the stability of this protein in normoxia. Akt/PKB also activates forkhead box O3a (FOXO3a), mammalian target of rapamycin (mTOR), and nuclear factor κB (NF-κB). Activation of mTOR leads to migration and an increase in CXCL8/IL-8 and VEGF expression. On the other hand, activation of NF-κB increases the expression of cytokines, such as tumor necrosis factor *α* (TNF-α) and CXCL16.

**Figure 3 ijms-22-03490-f003:**
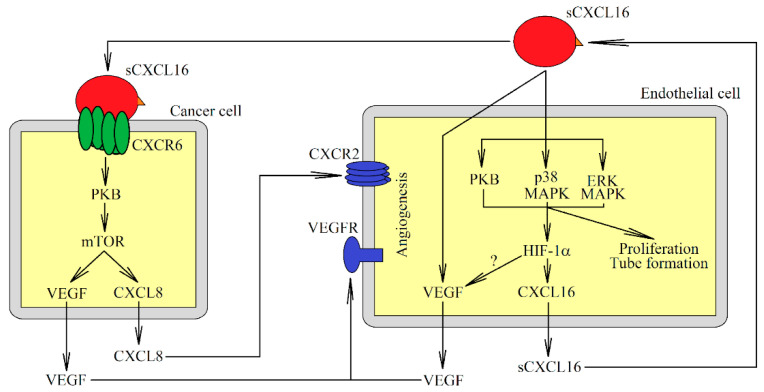
The role of sCXCL16 in angiogenesis. By activating CXCR6, sCXCL16 increases the expression of pro-angiogenic factors such as VEGF and CXCL8/IL-8 in the tumor cell. Also, sCXCL16 acts on endothelial cells and increases VEGF expression and proliferation, and can cause tube formation of these cells. sCXCL16 also increases the level of HIF-1α protein in endothelial cells, which leads to an increase in the expression of CXCL16 itself.

**Figure 4 ijms-22-03490-f004:**
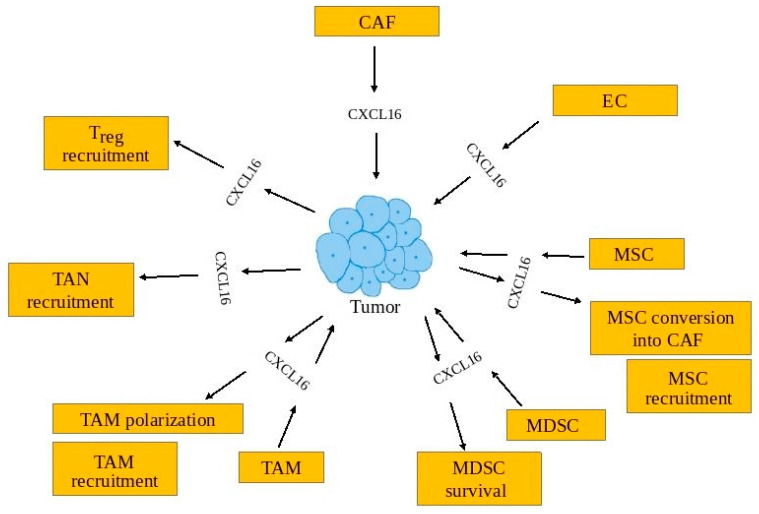
In the tumor, CXCL16-mediated intercellular communication creates multiple relationships between various cells. CXCL16 is produced not only by tumors but by tumor-associated cells, including CAF, EC, MSC, MDSC, and TAM. CXCL16 also influences tumor-associated cells. It causes recruitment of MSC, TAM, TAN, and T_reg_ into the tumor niche. CXCL16 also causes TAM polarization, MDSC survival, and MSC conversion into CAF.

**Table 1 ijms-22-03490-t001:** Activities of the two different forms of CXCL16.

Ligand	Receptor	Activated Signaling Pathways	Physiological Significance	References
sCXCL16	CXCR6	G protein-coupled receptor, PKB, ERK MAPK, calcium mobilization	Proliferation, migration, fibrosis, VEGF and CXCL8/IL-8 expression	[7,12,24,25,26,27,28,29,30,31,32,33]
sCXCL16	mCXCL16	insensitive to pertussis toxin, ERK MAPK, PKB	Proliferation, apoptosis resistance	[21,23]
CXCR6	mCXCL16	insensitive to pertussis toxin, ERK MAPK	Migration but not proliferation	[22,23]
mCXCL16	CXCR6		Cell adhesion	[12,20]

**Table 2 ijms-22-03490-t002:** The importance of tumor-associated cells in the functioning of the CXCL16→CXCR6 axis.

Cell Type	Impact on Recruitment to the Cancer Niche	CXCL16 Expression	Cellular Effect	References
Astrocytes		X		[134]
Cancer-associated fibroblasts (CAF)		X		[76,135]
Endothelial cells (EC)		X		[74,76,120,134,136]
Mesenchymal stem cells (MSC)	X	X	Conversion into CAF	[94,95,137,138,139]
Microglia		X	Cause anti-inflammatory phenotype	[74,99,136,140]
Myeloid-derived suppressor cells (MDSC)		X	Survival of MDSC	[96,141,142]
Tumor-associated macrophages (TAM)	X	X	Polarization into M2 macrophage subset	[74,76,114,135,136,143,144,145,146]
Tumor-associated neutrophils (TAN)	X			[32,147]
Regulatory T cells (T_reg_)	X		Increase T_reg_ growth at <0.3 ng/mL	[85,107,148,149]

**Table 3 ijms-22-03490-t003:** Overall survival of patients with high tumor or serum CXCL16 expression.

Type of Cancer	Number of Patients	Overall Survival with Elevated Amounts of CXCL16	Location	References
Bladder cancer	155	--	tumor	[177]
Cervical cancer	60	↓	tumor, *p* = 0.089	[125]
Colon cancer	121	↓	regional lymph nodes	[171]
Colorectal cancer	58	↑	tumor	[153]
Colorectal cancer	314	↓	serum	[98]
Colorectal cancer	142	↓	tumor	[172]
Ewing sarcoma family tumor	61	↓	tumor	[126]
Gastric carcinoma	359	↑	tumor	[173]
Gastrointestinal stromal tumor	43	↓	tumor	[107]
Gastrointestinal stromal tumor	43	↓	serum	[107]
Lung cancer (non-small cell lung cancer)	58	--	tumor	[179]
Lung cancer (non-small cell lung cancer)	58	--	serum	[179]
Lung cancer (non-small cell lung cancer)	301	↑	tumor	[78]
Lung cancer (non-small cell lung cancer)	40	--	serum	[181]
Lung cancer	56	↓	tumor	[28]
Ovarian carcinoma	56	↓	tumor	[128]
Ovarian cancer	273	--	tumor	[180]
Ovarian cancer	118	↓	serum	[180]
Prostate cancer	470	↓	tumor	[108]
Renal cell carcinoma	104	↑	tumor	[14]
Thyroid cancer (papillary thyroid cancer)	492	--	tumor, from TCGA dataset	[145]

↑-better overall survival; ↓-poorer overall survival; ---no correlation.

**Table 4 ijms-22-03490-t004:** Effect of CXCL16 or CXCR6 expression in the tumor on patient overall survival according to “The Human Protein Atlas” (https://www.proteinatlas.org, accessed on 15 January 2021) [182,183].

Type of Cancer	Overall Survival for Increased CXCL16 Expression in the Tumor	Overall Survival for Increased Expression of CXCR6 in the Tumor
Glioma	↓*p* = 0.094	↓*p* = 0.078
Thyroid cancer	↑	↑
Lung cancer	--	↑
Colorectal cancer	--	↑
Head and neck cancer	--	↑
Stomach cancer	↓	↑
Liver cancer	↓	↑
Pancreatic cancer	↑	--
Renal cancer	↑	↓
Urothelial cancer	--	↑
Prostate cancer	--	--
Testis cancer	↓	↓
Breast cancer	↑	↑
Cervical cancer	↑*p* = 0.052	↑
Endometrial cancer	↓	↑
Ovarian cancer	--	↑
Melanoma	--	↑

↑ blue background-better overall survival; ↓ red background–poorer overall survival; --–no correlation.

**Table 5 ijms-22-03490-t005:** Overall survival of patients with high tumor CXCR6 expression.

Type of Cancer	Number of Patients	Overall Survival for An Increased Amount of CXCR6	Comments	References
Bladder cancer	155	--		[177]
Cervical cancer	60	↓		[125]
Ewing sarcoma family tumor	61	↓		[126]
Gastric cancer	352	↓		[102]
Gastrointestinal stromal tumor	43	↓		[107]
Hepatocellular carcinoma	240	↓	*p* = 0.064	[32]
Lung cancer (non-small cell lung cancer)	58	--		[179]
Ovarian carcinoma	56	--		[128]
Ovarian cancer	268	--		[180]
Pancreatic ductal adenocarcinoma	112	↑	Early stage of pancreatic ductal adenocarcinoma cases, from TCGA dataset	[184]
Prostate cancer	476	↓		[108]
Renal cell carcinoma (clear cell renal cell carcinoma)	239	↓		[186]
Renal cell carcinoma	104	--		[14]
Thyroid cancer (papillary thyroid cancer)	136	--		[114]

↑-better overall survival; ↓-poorer overall survival; --–no correlation.

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
