# Peer review of "The Role of CXCL16 in the Pathogenesis of Cancer and Other Diseases"

_ijms, 2021, doi:10.3390/ijms22073490_

Round 1

Reviewer 1 Report

In this review, the authors present a comprehensive overview of the multiple roles of CXCL16 in tumors and other diseases. This topic is of interest and the literature evaluated complete and appropriate.

The tables and figure are informative and thorough, with some minor defects:

  • It would be easier for the reader to have more reference figures, for example a schematic drawing of the interaction between cancer cells and other cells of the microenvironment besides endothelial cells would be useful.
  • Table 1, I would suggest to add references.
  • Table 2, the references are not in the journal format.
  • Table 3, same as Table 1.
  • Table 4, same as Table 2.

The text is more problematic. Some specific points might be cited; however, this Reviewer feels that an appropriate reorganization and rephrasing of the whole text would be necessary to be able to judge the single points. In particular:

  • The parts on non-cancer diseases are off-topic and should either be deleted or just mentioned in the general background paragraphs.
  • There is a lot of redundancy and most information could be grouped to gain clarity.
  • Overall, the structure of this review looks like a list of results from many papers, so it is difficult to follow the logical flow. A suggestion is to try and organize this material differently. For example, the link between inflammation and CXCL16 could be evaluated in the different settings. But this is only a suggestion, since there are many alternative ways to organize the flow of speech.
  • The English needs a substantial revision.

Author Response

Review 1

In this review, the authors present a comprehensive overview of the multiple roles of CXCL16 in tumors and other diseases. This topic is of interest and the literature evaluated complete and appropriate.

The tables and figure are informative and thorough, with some minor defects:

  • It would be easier for the reader to have more reference figures, for example a schematic drawing of the interaction between cancer cells and other cells of the microenvironment besides endothelial cells would be useful.
  • Table 1, I would suggest to add references.
  • Table 2, the references are not in the journal format.
  • Table 3, same as Table 1.
  • Table 4, same as Table 2.

Tables have been supplemented according to the reviewer's guidelines. Table 3 includes a citation in its title.

The text is more problematic. Some specific points might be cited; however, this Reviewer feels that an appropriate reorganization and rephrasing of the whole text would be necessary to be able to judge the single points. In particular:

  • The parts on non-cancer diseases are off-topic and should either be deleted or just mentioned in the general background paragraphs.

  • The sections on the involvement of CXCL16 in noncancer diseases have been deleted from the initial subsections. Now it is mentioned are only in one part of the article

  • There is a lot of redundancy and most information could be grouped to gain clarity.

  • Overall, the structure of this review looks like a list of results from many papers, so it is difficult to follow the logical flow. A suggestion is to try and organize this material differently. For example, the link between inflammation and CXCL16 could be evaluated in the different settings. But this is only a suggestion, since there are many alternative ways to organize the flow of speech.

  • The English needs a substantial revision.

The text has been re-checked and corrected by the translator and native speaker

Reviewer 2 Report

This is a well organized review describing the role of CXCL16 a chemoattract participating in the potential initiation/progression of a number of diseases.  The authors describe these diseases, but mainly focus on its role in cancer.  The only comments I have is that there are a number of English sentence structure that need cleaned up prior to publication.  

Author Response

Review 2

This is a well organized review describing the role of CXCL16 a chemoattract participating in the potential initiation/progression of a number of diseases.  The authors describe these diseases, but mainly focus on its role in cancer.  The only comments I have is that there are a number of English sentence structure that need cleaned up prior to publication.  

The text has been re-checked and corrected by a translator and native speaker

Reviewer 3 Report

This manuscript, written by Dr Jan Korbecki et al, with the title of "The role of CXCL16 in the pathogenesis of cancer and otehr diseases", is a review type manuscript that focuses on the pathological role of CXCL16 and CXCR6 in cancer and other diseases.
CXCL16 is a chemokine that has many roles, but that is involved in chemotaxis, positive regulation of cell growth and migration, it is expressed on macrophages and tumoral cells, and it is also relevant in inflammatory conditions as well as other diseases such as atherosclerosis. The receptor of CXCL16 is the CXCR6, which is known for being used as a coreceptor by SIVs and some strains of HIV-2/1, but also have a role in cell chemotaxis, immune response, inflammation, etc. Therefore, the authors are making a review of an relevant marker.
After a quick introduction, the authors provides background information about CXCL16/CXCR6, and describes the membrane and soluble forms of CXCL16 and the membrane form of CXCR6. In this part, the intracellular signaling pathways are also described. Then, the authors focuses in non-neoplastic disease such as atherosclerosis, nonalcoholic fatty liver disease (NAFLD), and somes sentences for graft-vs-host disease (GVHD), fibrosis, inflammatory bowel disease and endometriosis. After that, the revision focuses on cancer (expression, cell proliferation, migration, metastasis), the immune tumoral microenvironment (angiogenesis, the different cellular components of the microenvironment including the relevant tumor-associated macrophages and tumor-infiltrating lymphocytes), the CXCL16-CXCR6 axis (expression and prognosis of the patients and anti-cancer therapy).
This manuscript is well written, it is easy to read although from a personal point of view sometimes I felt it had a lot of information that made me take a brake, the english is correct and standard, the figures and tables are clear, and there are enough references.
Before publishing this work, the authors may consider some minor recommendations that may help to improve the quality of this review text:1- From what I understood, there are two forms of CXCL16, the membrane mCXCL16 and the soluble sCXCL16. Do sCXCL16 can also interact with mCXCL16 as well as with CXCR6 receptor? The fact that these two markers can be expressed by multiple types of cells in different conditions can be confusing. A table may help to pinpoint the most relevant points.
2- In this manucript there is no information about the genomic alterations of these two markers. According to the data that can be found in the cBioPortal, not much changes are found for CXCL16 across the different types of cancers, only deep deletion is significantly found for undifferentiated stomach adenocarcinoma (around 8%) and neuroepithelial tumor (3%); for melanoma there are multiple changes (mutation, amplification, deletion) but also in low percetanges (around 3%). In case of CXCR6, the frequency of changes also seem low, around 4% for endometrial carcinoma, melanoma and sarcoma (all a mixture of mutation, amplification and deletion). It may be useful if the authors could explore the genomic/mutational/etc. landscape for these two markers and include some information in the review if interesting findings are found.
3- The authors have included data from the human protein atlas. I think that this webpage is very useful and I have used it myself. But I have found that some reviewers do not like you to cite it directly, and ask to back the information with research manuscripts, if available. In addition, some reviewers also do not like the use of word "prognosis" too much, and if avaiable ask to specify if it was overall survival, progression-free survival, etc. A note could we added in the tables.
4- After reading all the text, I was surprised not to find a final conclusion. Due to the fact that it is a very complex issue, I think that the authors could write a final conclusion, with they most important finding highlighted, and possibly a table as well.
5- The authors could write some sentences about HIV and immunedeficiencies, if relevant data is available.

Author Response

Review 3

  1. From what I understood, there are two forms of CXCL16, the membrane mCXCL16 and the soluble sCXCL16. Do sCXCL16 can also interact with mCXCL16 as well as with CXCR6 receptor? The fact that these two markers can be expressed by multiple types of cells in different conditions can be confusing. A table may help to pinpoint the most relevant points.Table has been added
  2.  
  3.  
  4. In this manucript there is no information about the genomic alterations of these two markers. According to the data that can be found in the cBioPortal, not much changes are found for CXCL16 across the different types of cancers, only deep deletion is significantly found for undifferentiated stomach adenocarcinoma (around 8%) and neuroepithelial tumor (3%); for melanoma there are multiple changes (mutation, amplification, deletion) but also in low percetanges (around 3%). In case of CXCR6, the frequency of changes also seem low, around 4% for endometrial carcinoma, melanoma and sarcoma (all a mixture of mutation, amplification and deletion). It may be useful if the authors could explore the genomic/mutational/etc. landscape for these two markers and include some information in the review if interesting findings are found.Information about genetic changes in CXCL16 and CXCR6 genes has been added 

  5. 3- The authors have included data from the human protein atlas. I think that this webpage is very useful and I have used it myself. But I have found that some reviewers do not like you to cite it directly, and ask to back the information with research manuscripts, if available. In addition, some reviewers also do not like the use of word "prognosis" too much, and if avaiable ask to specify if it was overall survival, progression-free survival, etc. A note could we added in the tables.
  6.  

The text has been revised according to the reviewer's recommendations. Nevertheless, the aim of our review is to completely discuss the importance of CXCL16 in cancer processes. For this reason, we have included data showing the association of CXCL16 and CXCR6 expression on overall survival. We have included these data from available articles on PubMed and data from "the human protein atlas". We can remove the data from "the human protein atlas" but this will greatly reduce the value of our article.

  1. After reading all the text, I was surprised not to find a final conclusion. Due to the fact that it is a very complex issue, I think that the authors could write a final conclusion, with they most important finding highlighted, and possibly a table as well.

Conclusions have been added.

5- The authors could write some sentences about HIV and immunedeficiencies, if relevant data is available.

The passage about HIV has been added.

Round 2

Reviewer 1 Report

This revised version is improved in terms of both clarity and English language. I have a few minor comments:

  1. A further language check is recommended, for example regarding the terms "both" and "in turn", which are often used inappropriately.
  2. Another suggestion is to double-check that the two forms of the CXCL16 protein are correctly mentioned throughout the manuscript. Since mCXRC16 and sCXCL16 have a different function and, consequently, different biological effects, it is pivotal not to get confused. Also, in some cases CXCL16 is mentioned without reference to the protein form and the reason for this choice should be explained.
  3. What does the sentence “Apart from mCXCL16, we know only one another receptor for CXCL16, namely CXCR6” mean? (page 4, Chapter 3. CXCR6: background information). The reason why mCXCL16 is considered a receptor for CXCL16 itself should be explained.
  4. There are some contradictory statements, for example regarding the expression of CXCR6 in macrophages: compare “CXCR6 is not expressed on DC, macrophages, monocytes or neutrophils [6,32]” with “CXCR6 expression is higher in the M1 than in the M2 macrophage subset [34]” (page 4, Chapter 3. CXCR6: background information). While it is normal that different studies generate different results, this should be stated / explained in the text.
  5. There are some formatting inaccuracies throughout the manuscript.

Author Response

  1. A further language check is recommended, for example regarding the terms "both" and "in turn", which are often used inappropriately.

The paper has been re-read and corrected for errors, including the inappropriate use of “both” and “in turn”.

  1. Another suggestion is to double-check that the two forms of the CXCL16 protein are correctly mentioned throughout the manuscript. Since mCXRC16 and sCXCL16 have a different function and, consequently, different biological effects, it is pivotal not to get confused. Also, in some cases CXCL16 is mentioned without reference to the protein form and the reason for this choice should be explained.

During the writing of the paper we paid special attention to which of the two forms of CXCL16 was studied in the available publications. However, very often in experimental work researchers do not pay attention to which form has the demonstrated effect. Very often they just increase or decrease the expression of the CXCL16 gene, or change the expression of two forms of the described protein at the same time. Then they test certain parameters of the experiment. For this reason, in our review article when writing about CXCL16 we mean an unspecified form which maybe refers to both forms of CXCL16. When writing about mCXCL16 and sCXCL16, we always mean the specific form of CXCL16.

(we have included this thought in the paper).

  1. What does the sentence “Apart from mCXCL16, we know only one another receptor for CXCL16, namely CXCR6” mean? (page 4, Chapter 3. CXCR6: background information). The reason why mCXCL16 is considered a receptor for CXCL16 itself should be explained.

This is described as "reverse signaling" in the paragraph on the transduction of signal from mCXCL16

  1. There are some contradictory statements, for example regarding the expression of CXCR6 in macrophages: compare “CXCR6 is not expressed on DC, macrophages, monocytes or neutrophils [6,32]” with “CXCR6 expression is higher in the M1 than in the M2 macrophage subset [34]” (page 4, Chapter 3. CXCR6: background information). While it is normal that different studies generate different results, this should be stated / explained in the text.

These are two experimental papers with opposite results. We think this is caused by the difference in the research model. The first paper tested CXCR6 expression in vivo, while the second paper tested CXCR6 expression in vitro on monocytes treated with appropriate agents. The relevant passage has been added to the text.

  1. There are some formatting inaccuracies throughout the manuscript.

The text has been re-checked with regard to formatting.
